# Frontotemporal Lobar Degeneration Case with an N-Terminal *TUBA4A* Mutation Exhibits Reduced TUBA4A Levels in the Brain and TDP-43 Pathology

**DOI:** 10.3390/biom12030440

**Published:** 2022-03-12

**Authors:** Evelien Van Schoor, Mathieu Vandenbulcke, Valérie Bercier, Rik Vandenberghe, Julie van der Zee, Christine Van Broeckhoven, Markus Otto, Bernard Hanseeuw, Philip Van Damme, Ludo Van Den Bosch, Dietmar Rudolf Thal

**Affiliations:** 1Laboratory of Neuropathology, Department of Imaging and Pathology, Leuven Brain Institute (LBI), KU Leuven (University of Leuven), 3000 Leuven, Belgium; 2Laboratory of Neurobiology, Department of Neurosciences, Leuven Brain Institute (LBI), KU Leuven (University of Leuven), 3000 Leuven, Belgium; valerie.bercier@kuleuven.be (V.B.); philip.vandamme@kuleuven.be (P.V.D.); ludo.vandenbosch@kuleuven.be (L.V.D.B.); 3Center for Brain & Disease Research, VIB, 3000 Leuven, Belgium; 4Department of Geriatric Psychiatry, University Hospitals Leuven, 3000 Leuven, Belgium; mathieu.vandenbulcke@uzleuven.be; 5Laboratory of Cognitive Neurology, Department of Neurosciences, KU Leuven (University of Leuven), 3000 Leuven, Belgium; rik.vandenberghe@uzleuven.be; 6Department of Neurology, University Hospitals Leuven, 3000 Leuven, Belgium; 7Neurodegenerative Brain Diseases, Center for Molecular Neurology, VIB, 2610 Antwerp, Belgium; julie.vanderzee@uantwerpen.vib.be (J.v.d.Z.); christine.vanbroeckhoven@uantwerpen.vib.be (C.V.B.); 8Department of Biomedical Sciences, University of Antwerp, 2000 Antwerp, Belgium; 9Department of Neurology, Ulm University, 89081 Ulm, Germany; markus.otto@uk-halle.de; 10Department of Neurology, University Hospital Halle (Saale), Martin Luther University Halle-Wittenberg, 06108 Halle (Saale), Germany; 11UC Louvain and Department of Neurology, Institute of Neurosciences, University Hospital Saint-Luc, 1200 Brussels, Belgium; bernard.hanseeuw@saintluc.uclouvain.be; 12Department of Pathology, University Hospitals Leuven, 3000 Leuven, Belgium

**Keywords:** frontotemporal dementia, TUBA4A, transactive response DNA-binding protein 43 kD

## Abstract

Recently, disease-associated variants of the *TUBA4A* gene were identified in patients with amyotrophic lateral sclerosis (ALS) and frontotemporal dementia (FTD). Here, we present the neuropathological report of a patient with the semantic variant of primary progressive aphasia with a family history of Parkinsonism, harboring a novel frameshift mutation c.187del (p.Arg64Glyfs*90) in *TUBA4A*. Immunohistochemistry showed abundant TAR DNA-binding protein 43 kDa (TDP-43) dystrophic neurite pathology in the frontal and temporal cortex and the dentate gyrus of the hippocampus, consistent with frontotemporal lobar degeneration (FTLD). The observed pathology pattern fitted best with that of FTLD-TDP Type C. qPCR showed the presence of mutant *TUBA4A* mRNA. However, no truncated TUBA4A was detected at the protein level. A decrease in total *TUBA4A* mRNA and protein levels suggests loss-of-function as a potential pathogenic mechanism. This report strengthens the idea that N-terminal *TUBA4A* mutations are associated with FTLD-TDP. These N-terminal mutations possibly exert their pathogenic effects through haploinsufficiency, contrary to C-terminal *TUBA4A* mutations which are thought to disturb the microtubule network via a dominant-negative mechanism.

## 1. Introduction

Frontotemporal dementia (FTD) is a type of early-onset dementia characterized by atrophy of the frontal and temporal lobes. Clinical symptoms can present as behavioral changes, cognitive deficits and language dysfunction [1]. Based on these clinical criteria, different phenotypes can be distinguished, including the behavioral variant (bvFTD), the nonfluent variant of primary progressive aphasia (nfPPA) and the semantic variant of primary progressive aphasia (svPPA) [2,3]. FTD can also co-exist with atypical parkinsonian disorders, such as progressive supranuclear palsy (PSP) and corticobasal syndrome (CBS), and with amyotrophic lateral sclerosis (ALS) [1]. The pathology underlying FTD is referred to as frontotemporal lobar degeneration (FTLD), which can present as FTLD-TDP, FTLD-FUS or FTLD-tau, reflecting the protein that aggregates in affected brain regions [4]. Up to 40% of FTD cases present with a family history of dementia or psychiatric illness, although only 10% can be explained by known gene mutations. The most frequently affected genes are *C9orf72*, *MAPT* and *GRN* [5]. The *C9orf72* repeat expansion is the most common cause of FTD as well as ALS, stressing the overlap between both diseases. More recently, variants in other genes were identified in ALS/FTD patients, including variants in *TUBA4A* [6,7,8]. The *TUBA4A* gene encodes the alpha tubulin 4A protein, one of nine human α-tubulins, which polymerize with β-tubulin to form the structural subunits of microtubules. One of the major functions of microtubules in the central nervous system is the regulation of transport along the axon [9]. *TUBA4A* is ubiquitously expressed in all cell types, but has its highest levels of expression in the brain. The expression of *TUBA4A* also increases over time, possibly explaining why mutations in this gene cause advanced age disease phenotypes [10]. Disease-associated variants of the *TUBA4A* gene were mainly identified in patients with familial ALS, some of whom had signs of cognitive impairment [6]. Most of these mutations occur in the C-terminal end of the protein, which is important for its interaction with other tubulin subunits and associated proteins, such as kinesin [11]. More recently, however, mutations in the N-terminal region of *TUBA4A* were observed in patients with FTD without motor neuron disease [7,12]. In this report, we present the neuropathological *post-mortem* analysis of an FTD patient presenting with the semantic variant (svPPA) and an R64Gfs*90 *TUBA4A* mutation, and suggest reduction of TUBA4A protein levels as a potential pathogenic mechanism.

## 2. Materials and Methods

### 2.1. Human Autopsy Cases

Central nervous system (CNS) tissue was collected in the UZ Leuven brain biobank in accordance with the ethics review board upon written informed consent. Only autopsy of the brain with adjacent upper cervical spinal cord was granted. In addition to the FTLD-TDP case with the R64Gfs*90 *TUBA4A* mutation (case n° 1), four FTLD-TDP cases and six non-neurodegenerative controls were included in this study (Appendix A). For these four FTLD-TDP cases, genetic testing was performed for *TUBA4A*, *GRN*, *MAPT*, *VCP*, *TARDBP*, *FUS*, *SOD1*, *TBK1*, *PSEN1*, *PSEN2*, *APP*, and *C9orf72*, which was negative for cases 2, 3, and 4. One FTLD-TDP patient carried a *GRN* mutation (IVS1+5G>C; case n° 5) (Appendix A). The diagnosis of FTD was based on clinical assessment [2,3] and confirmed by ^18^F-FDG-PET and MRI (Figure 1). The diagnosis of FTLD-TDP was pathologically confirmed by assessment of the pTDP-43 pathology. The R64Gfs*90 *TUBA4A* mutation was identified by exonic sequencing as described in detail in Perrone et al. [7]. Mutations in other ALS/FTD-related genes were excluded for this patient (case n° 1) (i.e., *C9orf72*, *MAPT*, *VCP*, *TARDBP*, *FUS*, *SOD1*, *TBK1*, *ATXN2*, *UBQLN2*, *SQSTM1*, and *TREM2*). For the control cases used in this study, genetic data were not available.

### 2.2. RT-qPCR

RNA was extracted from frontal cortex and temporal cortex of the R64Gfs*90 *TUBA4A* mutation case, and from frontal cortex of control cases (*n* = 3) using the RNeasy Mini Kit (Qiagen, Hilden, Germany). RNA integrity numbers (RIN) were determined using the Agilent 6000 Bioanalyzer Nano or Pico chip (Agilent Technologies, Santa Clara, CA, USA). RIN values are shown in Appendix A. cDNA was generated using the GoScript^TM^ Reverse Transcriptase kit (Promega, Madison, WI, USA). cDNA was then used as template for qPCR using GoTaq Probe qPCR reagents (Promega, Madison, WI, USA). The qPCR was performed on the CFX96 RT system (Biorad, Hercules, CA, USA) using a 96-well plate in technical triplicates. All signals were normalized to GAPDH (Integrated DNA Technologies, Leuven, Belgium). The 2^−ΔΔCt^ method was used to calculate the fold change of RNA level compared to control samples.

The following primers were used for *TUBA4A*:

Forward *TUBA4A* primer: 5′ GAC TCC TTC ACC ACC TTC TTC 3′

Reverse *TUBA4A* primer: 5′ CGG ATC TCA TCA ATG ACC GTA G 3′

Two different locked nucleic acid (LNA) probes were used in combination and worked through competitive binding to distinguish the mutant from the wild-type *TUBA4A*:

Mutant LNA probe: 5′ FAM/A+CG+T+A+C CC+G G/3IABkFQ 3′

Wild-type LNA probe: 5′ HEX/CGT+A+C+C CC+G G/3IABkFQ 3′

### 2.3. Immunohistochemistry

Histological examination of the *TUBA4A* mutation case was performed on 5 μm-thick sections cut from formalin-fixed, paraffin-embedded tissue of frontal cortex, cingulate gyrus, parietal cortex, temporal cortex, occipital cortex, hippocampus, entorhinal cortex, hypothalamus, basal ganglia, amygdala, thalamus, midbrain, pons, medulla oblongata, cerebellum, cervical spinal cord, and pre/postcentral cortex. Frontal and temporal cortex from sporadic FTLD-TDP type C cases and non-neurodegenerative controls were stained in parallel. pTDP-43 (1:5000, TIP-PTD-P02, Cosmo Bio, Tokyo, Japan) stainings were performed automatically by means of the BOND-MAX automated IHC/ISH Stainer (Leica Biosystems, Wetzlar, Germany) using the Bond Polymer Refine Detection kit (DS9800, Leica Biosystems, Wetzlar, Germany). Immunohistochemistry for C-t TDP-43 (1:1000, 12892-1-AP, Protein Tech, Manchester, UK), N-t TDP-43 (1:400, ARP38941_T100, Aviva Systems Biology, San Diego, CA, USA), TUBA4A (1:100, AP13535b, Abgent, San Diego, CA, USA) and CD68 (1:100, M0814, Dako, Agilent Technologies, Santa Clara, CA, USA) was performed manually. Afterwards, hematoxylin counterstaining was performed in the Leica ST5010 Autostainer XL (Leica Biosystems, Wetzlar, Germany). To exclude Alzheimer’s disease (AD), other tauopathies and Lewy Body Dementia (LBD) [13,14,15], the hippocampus, entorhinal cortex, and occipital cortex were immunostained with antibodies against Aβ (1:5000, 39220 clone 4G8, BioLegend, San Diego, CA, USA) and phospho-tau (1:1000, clone AT8, Pierce-Endogen, Woburn, MA, USA), and the medulla oblongata was immunostained using an antibody targeting α-synuclein (1:4000, clone 5G4, Millipore, Burlington, MA, USA).

### 2.4. Immunofluorescence

Immunofluorescence double-labeling of 5μm sections of the frontal cortex of the R64Gfs*90 *TUBA4A* mutation case was performed for TUBA4A (1:100, AP13535b, Abgent, San Diego, CA, USA) and TDP-43 (1:1000, 60019-2-Ig, Protein Tech, Manchester, UK), labeled with goat anti-rabbit Cy3 (1:100, 111-165-144, Jackson ImmunoResearch, West Grove, PA, USA) and goat anti-mouse Cy2 (1:50, 115-225-146, Jackson ImmunoResearch, West Grove, PA, USA). Slides were mounted using ProLong Gold Antifade Mountant (Thermo Fisher Scientific, Rockford, IL, USA).

### 2.5. Protein Extraction

Human autopsy brains from the R64Gfs*90 *TUBA4A* mutation case, four FTLD-TDP cases, and five controls were used in this study. The right hemispheres were cut in approx. 1 cm slabs and frozen at −80 °C. 50 mg of the brain tissue was weighed and mechanically homogenized in 0.5 mL 2% SDS in TBS (Tris-buffered saline) + Nuclease (Pierce^TM^ Universal Nuclease, Thermo Fisher Scientific, Rockford, IL, USA) + a cocktail of protease/phosphatase inhibitors (Halt, Thermo Fisher Scientific, Rockford, IL, USA) using a micropestle. Samples were sonicated, followed by a centrifugation at 13,000× *g* for 30 min. The resulting supernatant was used as the total fraction. Protein concentrations of the different fractions were determined using the Pierce BCA Protein Assay Kit (Thermo Fisher Scientific, Rockford, IL, USA).

### 2.6. Western Blotting

Samples (10 μg) were loaded on a Bis-Tris 4–12% gradient SDS-PAGE (Invitrogen, Waltham, MA, USA) and transferred to a nitrocellulose membrane (Semidry transfer, Biorad, Hercules, CA, USA). Membranes were blocked with 5% milk. Primary antibodies were TUBA4A (1:10 000, ab228701, Abcam, Cambridge, UK and 1:8000, AP13535b, Abgent, San Diego, CA, USA), N-t TUBA4A (1:1000, ab96743, Abcam, Cambridge, UK), and anti-HA-tag (1:1000, 3724, Cell Signaling, Danvers, MA, USA). Secondary antibodies were goat anti-rabbit IgG-HRP and goat anti-mouse IgG-HRP (polyclonal, Dako, Agilent Technologies, Santa Clara, CA, USA). The blots were developed with PICO plus ECL reagent (Thermo Fisher Scientific, Rockford, IL, USA). Digital images were acquired using the Amersham Imager 600 (GE Healthcare, Chicago, IL, USA). GAPDH (1:10,000, AM4300, Thermo Fisher Scientific, Rockford, IL, USA) was used as a loading control. Band intensities were measured using ImageJ.

### 2.7. TUBA4A Constructs

Human wild-type and W407* *TUBA4A* FLAG-tagged encoding plasmids were received from J. Landers [6]. The W407* construct was used as a positive control, as this frameshift mutation is also predicted to lead to a truncated TUBA4A protein fragment. We introduced the R64Gfs*90 mutation in the wild-type *TUBA4A* plasmid by Gibson cloning using overlapping primers containing the c.187del to amplify the whole plasmid (forward primer: 5′-ACGTACCCGGGCAGTTTTTGTGGATCTGGAG-3′; reverse primer: 5′-ACTGCCCGGGTACGTGTTTTCCAGCACCAG-3′). A single clone was selected and correct insertion of the mutation (c.187del) was confirmed by sequencing. To produce mRNA, the plasmids were linearized by restriction digestion, transcribed with mMESSAGE mMACHINE T7 kit (Ambion, Huntingdon, UK) and the resulting mRNA was purified with the MEGAclear Kit (Ambion, Huntingdon, UK). The mRNA concentration was measured by spectrophotometry (Nanodrop, Thermo Fisher Scientific, Rockford, IL, USA). mRNA quality and length were verified by RNA gel electrophoresis.

### 2.8. Zebrafish Injections

One- to two-cell-stage zebrafish embryos from the AB strain were injected in the yolk sac with 300 ng/µL of mRNA. Injected embryos were raised in embryo medium and kept in a 28.5 °C incubator. For western blotting, fish were manually dechorionated with forceps at 6 h post fertilization. Only morphologically normal embryos were retained and homogenized in RIPA buffer supplemented with protease and phosphatase inhibitors (Sigma-Aldrich, St. Louis, MO, USA) using a micropestle on a hand-held rotor. After centrifugation (3 min, 10,000× *g* rpm), supernatant was collected and analyzed by western blotting.

## 3. Results

We present the autopsy case of a 60-year old man diagnosed at the age of 51 with the semantic variant of primary progressive aphasia. This was confirmed by MRI and ^18^F-FDG-PET, showing bilateral anterior temporal atrophy, which was more right-sided, as well as hypometabolism (Figure 1). The patient showed prosopagnosia and problems with word retrieval and word comprehension. Later on, the patient also developed behavioral changes, such as obsessive-compulsive behavior, increased appetite with loss of table manners, discrete disinhibition, loss of decorum, and emotional indifference. Additionally, the patient displayed an inexhaustible glabellar reflex and extrapyramidal symptoms such as a discrete stooped posture and a decreased arm swing. The patient did not show any signs of motor neuron disease. There was a family history of Parkinsonism on the paternal side (Figure 2). The father of the patient (Figure 2, II:1) was diagnosed with multi system atrophy (Parkinsonian type). The paternal grandmother (Figure 2, I:2) suffered from Parkinson’s disease since the age of 50. Additionally, a younger sibling of the patient was diagnosed with Parkinson’s disease (Figure 2, III:4). Exonic sequencing of the semantic dementia patient (Figure 2, III:2, red arrow) revealed a novel frameshift mutation c.187del (p.Arg64Glyfs*90) in exon 2 of the *TUBA4A* gene, which was absent in control individuals and public databases as previously published. Mutations in other genes causative for ALS/FTD were excluded (i.e., *C9orf72*, *MAPT*, *VCP*, *TARDBP*, *FUS*, *SOD1*, *TBK1*, *ATXN2*, *UBQLN2*, *SQSTM1* and *TREM2*) [7]. Genetic testing was not performed for any of the family members of the patient.

Autopsy was carried out in accordance with the UZ Leuven Ethics Committee with written informed consent. Frozen and formalin fixed tissue was stored in the UZ Leuven biobank. Macroscopically, the brain weighed 955 g and exhibited severe atrophy of the medial temporal lobe including the amygdala, enlarged ventricles and mild atherosclerosis of the circle of Willis. Microscopically, the presence of phosphorylated transactive response DNA-binding protein 43 kDa (pTDP-43)-positive dystrophic neurites and cytoplasmic inclusions in the frontal and temporal cortex confirmed the pathological diagnosis of frontotemporal lobar degeneration (FTLD), more specifically FTLD-TDP [16]. Additionally, neurofibrillary tangle (NFT) pathology (Braak-NFT stage I) [17] and amyloid plaques (Aβ phase I) [18] were present, indicative for an early stage of preclinical Alzheimer’s disease (AD) pathology. Precentral cortex pTDP-43 pathology was scarce, while the cervical spinal cord did not show any pTDP-43-positive lesions. Moderate microglia activation was present in pyramidal tracts in the medulla oblongata and precentral cortex as observed in CD68-stained sections. In the medulla oblongata, the degree of microglia activation was similar in all subregions and not accentuated in motor areas. No obvious pyramidal tract degeneration was detected in H&E stained sections. These observations suggest that the patient did not have apparent preclinical motor neuron disease. α-synuclein lesions were not detected. Extracellular melanine was observed in the substantia nigra. Mild cerebral amyloid angiopathy without capillary involvement was present, but no brain infarct or bleedings were observed.

In detail, immunohistochemistry (IHC) showed abundant pTDP-43 pathology primarily in the frontal and temporal cortex and the dentate gyrus of the hippocampus, consistent with FTLD-TDP. pTDP-43 pathology was widespread and reached the occipital cortex. The cerebellum and cervical spinal cord were devoid of pTDP-43 pathology (Appendix A). The observed pTDP-43 pathology consisted of abundant long thin, but also long and short thick dystrophic neurites and few cytoplasmic inclusions in all layers of the frontal and temporal cortex, with more prominent pathology in layers II and V (Figure 3a–h and Appendix A). This was also observed in the parietal and occipital cortex (Appendix A). The R64Gfs*90 *TUBA4A* mutation case showed more dystrophic neurites in the deeper layers of the frontal and temporal cortex and in the dentate gyrus, compared to typical FTLD-TDP Type C cases (Figure 3a–h; Appendix A) [19,20]. No intranuclear inclusions were observed. The white matter was virtually free of pTDP-43 pathology. Antibodies against the C- and N-terminus of TDP-43 confirmed the pattern of pTDP-43 pathology, indicating that the majority of the lesions consisted of phosphorylated and non-phosphorylated full-length TDP-43 (Appendix A). Accordingly, this R64Gfs*90 *TUBA4A* mutation case fits best with the FTLD-TDP Type C pattern (Appendix A) [6,7], because of the presence of abundant dystrophic neurites of various length and thickness and few neuronal cytoplasmic inclusions both in the superficial and deep layers of the cortex, and the clinical diagnosis of semantic dementia. Of notice, the substantial layer V pathology as seen in this case is not characteristic of FTLD-TDP Type C.

Next, we evaluated whether the R64Gfs*90 *TUBA4A* mutation was associated with changes in TUBA4A protein expression and distribution in the brain of the patient. TUBA4A IHC of the frontal and temporal cortex showed an altered neurite architecture (Appendix A). These thick neurites positive for TUBA4A did not co-localize with pTDP-43 dystrophic neurites (Appendix A). No TUBA4A aggregates were identified.

qPCR analysis using competitive probes against the WT (wild-type) and R64Gfs*90 mutant *TUBA4A* showed the presence of mutant *TUBA4A* RNA in the R64Gfs*90 *TUBA4A* mutation case (Table 1). As the R64Gfs*90 *TUBA4A* mutation leads to a premature termination codon in exon 4, 90 amino acids from the frameshift, a fragmented protein product could be expected around 17 kDa (Figure 4a; arrow). However, western blot analysis on R64Gfs*90 *TUBA4A* mutation case brain lysates using an antibody directed against the N-terminal part of the TUBA4A protein did not reveal any TUBA4A fragment produced by the frameshift mutation (Figure 4b). This suggests that the mutant R64Gfs*90 *TUBA4A* is degraded at the RNA or protein level. This was confirmed by the injection of the R64Gfs*90 mutant *TUBA4A* mRNA in zebrafish embryos, showing absence of a TUBA4A fragment on western blot, in contrast to wild-type and W407* mutant TUBA4A (Figure 4c). Furthermore, qPCR analysis showed a reduction in WT *TUBA4A* mRNA compared to control cases, which can be expected from a frameshift mutation (Table 1). This was reflected at the protein level, as the total level of TUBA4A was decreased in affected brain regions of the R64Gfs*90 *TUBA4A* mutation case compared to control and other FTLD-TDP cases (Figure 5a,b).

## 4. Discussion

In this study, we performed a neuropathological *post-mortem* analysis on a patient with an R64Gfs*90 *TUBA4A* mutation presenting with svPPA and underlying FTLD-TDP, without a motor neuron phenotype. Histopathological and biochemical analysis of frontal and temporal cortex fitted best with FTLD-TDP Type C, with prominent pTDP-43 lesions even in the deeper cortical layers. Western blot analysis did not show a TUBA4A protein fragment, which was confirmed by the injection of R64Gfs*90 *TUBA4A* mRNA in zebrafish. In contrast, injection of another ALS-related frameshift *TUBA4A* mutation (W407*) led to the production of a shortened protein product. On the other hand, we observed a reduction in wild-type *TUBA4A* mRNA as well as full-length TUBA4A protein levels in the R64Gfs*90 *TUBA4A* patient, pointing towards haploinsufficiency as a possible underlying pathogenic mechanism. However, we cannot fully exclude the possibility that a shortened TUBA4A protein product was not picked up by the TUBA4A N-terminal antibody used in the study, although in theory it should detect the conserved epitope upstream of the mutation located at amino acid 64.

Axonal transport defects have recently been reported to be involved in several neurodegenerative disorders [21,22,23,24,25]. Apart from microtubules consisting of α- and β-tubulin, motor proteins such as kinesin and dynein are also important players as they move cargoes along the microtubule scaffold and contribute to cytoskeleton stability and maintenance. In addition to *TUBA4A*, genetic mutations in motor proteins (e.g., *KIF5A*, *DCTN1*) have also been associated with ALS, further strengthening the hypothesis that alterations in proteins with an important function in cytoskeleton structure and dynamics are of major importance in ALS pathobiology [21,26]. Importantly, TUBA4A, unlike most other α-tubulin isotypes, does not contain a C-terminal tyrosine residue. In α-tubulin, this final tyrosine can be added or removed through the detyrosination-tyrosination cycle [27,28]. Detyrosination was shown to increase microtubule stability and is highly abundant in the axonal compartment. This suggests that sufficient expression of *TUBA4A* in the brain might be important for the formation of stable microtubules, such as in axons in neurons [9].

In 2014, several variants were identified in the C-terminal part of the *TUBA4A* gene in a cohort of ALS patients. These variants were reported to lead to classical spinal onset ALS, with, in some cases, FTD-like symptoms [6]. The C-terminal region of α-tubulin is important in its interaction with β-tubulin and microtubule-associated proteins (MAPs), such as dynein and kinesin [29]. Smith and colleagues showed that these ALS-related variants ineffectively formed tubulin dimers in vitro, and that they exhibited a decreased incorporation into protofilaments, possibly interfering with the microtubule network through a dominant-negative mechanism [6].

More recently, Mol et al., described another *TUBA4A* variant (R105C) in a family with different forms of dementia, among which bvFTD with prominent disinhibited behavior and parkinsonian-like gait disturbances [12]. One family member displayed unspecified dementia and comorbid Parkinsonism, and another relative was clinically diagnosed with Parkinson’s disease. None of the patients displayed ALS-like symptoms. *Post-mortem* analysis of two bvFTD cases showed decreased TUBA4A protein levels in the brain [12], which is in line with what we observed in the R64Gfs*90 *TUBA4A* patient presented in this paper. Another group reported a nonsense R79X *TUBA4A* mutation in a patient with Parkinson syndrome and nigropathy. Both siblings were also affected. They did not detect any R79X TUBA4A protein fragment and therefore suggested haploinsufficiency as a potential mechanism, although a decrease in full-length TUBA4A protein levels was not investigated in this patient [30].

Of notice, both the R105C [12] and R79X [30] *TUBA4A* mutations, as well as the R64Gfs*90 *TUBA4A* frameshift mutation reported here, localize in the N-terminal part of the *TUBA4A* gene (Figure 4a). The N-terminus is mainly important in protein folding and conformation [12]. However, mutations described in the N-terminal region of *TUBA4A* most likely act through a loss-of-function mechanism, as we and others showed that patients with these N-terminal *TUBA4A* mutations display a reduction in TUBA4A protein expression. Interestingly, the father and grandmother of the patient described here presented with multiple system atrophy (MSA) and Parkinson’s disease, and the R64Gfs*90 *TUBA4A* patient exhibited an inexhaustible glabellar reflex and stooped posture with decreased arm swing. TDP-43 pathology was observed in the caudate nucleus, putamen and globus pallidus of the R64Gfs*90 *TUBA4A* patient, indicating an involvement of the nigro-striatal system without a direct impact on the substantia nigra. Given the lack of α-synuclein pathology in our patient, extrapyramidal Parkinson symptoms such as stooped posture and decreased arm swing may mainly be related to the affection of the nigro-striatal system by TDP-43 pathology. This further supports the notion that variants in the N-terminal region of *TUBA4A* are more likely to be associated with FTD with extrapyramidal Parkinson-like symptoms, possibly exerting its pathogenic effects through a reduction in TUBA4A, whereas variants in the C-terminal region have mainly been associated with ALS, likely disrupting the microtubule network through a dominant-negative mechanism [6]. However, it remains unknown if these *TUBA4A* mutations are sufficient to develop ALS and/or FTD. Therefore, further studies assessing the impact of these mutations in different cell types in vitro and in vivo will help shed light on the pathogenicity and downstream effects, as well as on the selective vulnerability of neuronal populations.

Remarkably, some reports also showed downregulation of TUBA4A protein levels in familial as well as sporadic ALS patients in the absence of *TUBA4A* mutations, suggesting that alterations in the expression of *TUBA4A* could be of importance in sporadic ALS disease pathogenesis as well [31,32,33]. Helferich and colleagues described an miR-1825/*TBCB*/*TUBA4A* pathway, demonstrating that the reduced expression of an upstream miRNA can lead to a reduction in TUBA4A protein expression in ALS patients [31]. Future models for mutant and downregulated TUBA4A are needed to better understand its mechanistic role in ALS and FTLD.

## 5. Conclusions

In conclusion, this report supports the importance of N-terminal *TUBA4A* mutations in FTLD-TDP without ALS-like symptoms. We and other groups observed reduced TUBA4A protein levels in affected FTLD-TDP patients, pointing towards *TUBA4A* haploinsufficiency as a potential pathogenic mechanism. Functional studies will be essential in the further elucidation of the pathogenic mechanism of both C- and N-terminal *TUBA4A* mutations. Overall, these results further emphasize the important role of cytoskeletal defects in FTLD and ALS pathobiology.

## Figures and Tables

**Figure 1 biomolecules-12-00440-f001:**
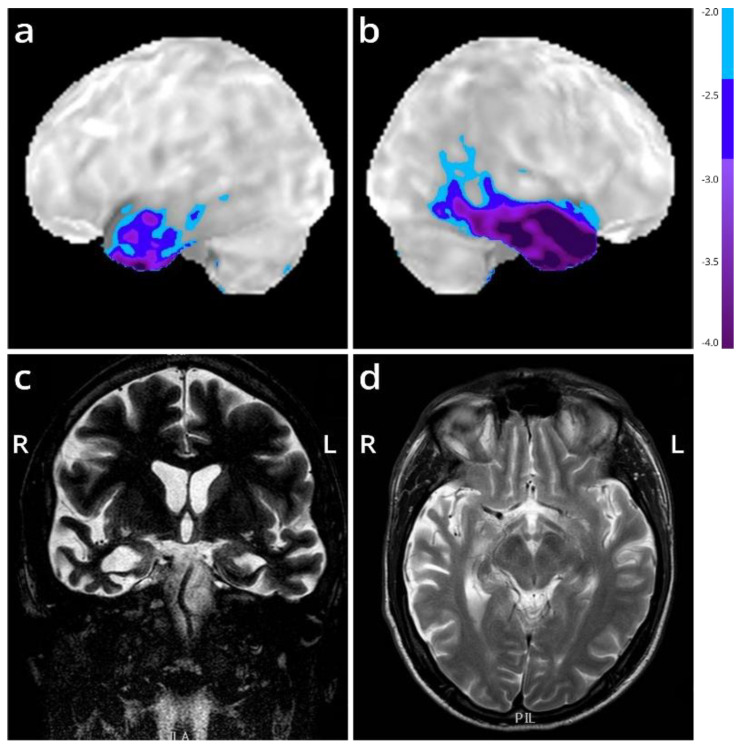
Imaging of the FTD patient with an R64Gfs*90 *TUBA4A* mutation. (**a**,**b**) Stereotactic surface projections showing lateral views of the left (**a**) and right (**b**) hemisphere of an ^18^F-FDG-PET of the patient, with areas of significant decrease of glucose metabolism superimposed. The scan shows a regional decrease in glucose metabolism in the anterior temporal lobes, more pronounced to the right compared to the left. Color-coding refers to Z-scores with respect to a dataset of normal control subjects. (**c**,**d**) MRI (T2-weighted) at the level of the temporal lobe (coronal in (**c**); horizontal in (**d**)) shows temporal lobar degeneration typical for FTLD. L = left; R = right.

**Figure 2 biomolecules-12-00440-f002:**
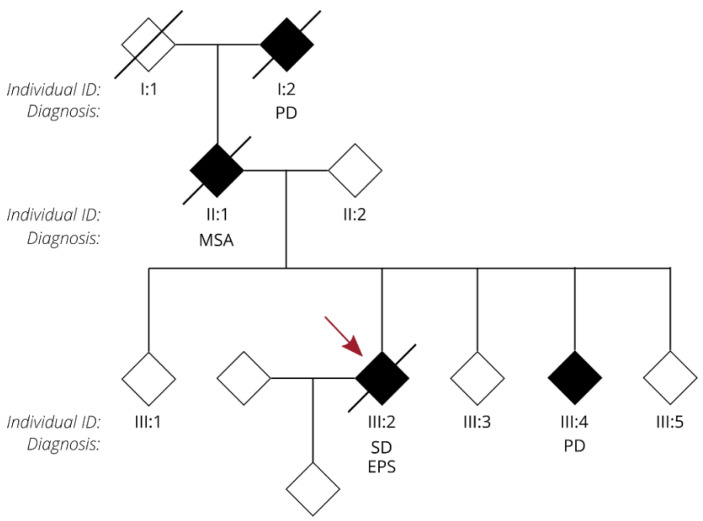
Pedigree of the family of the FTD patient with an R64Gfs*90 *TUBA4A* mutation. Diamond-shaped symbols were used for anonymity. Filled black symbols represent clinically affected patients. A diagonal line marks deceased patients. The individual ID and relevant clinical neurological diagnosis are mentioned for each patient. The FTD patient with an R64Gfs*90 *TUBA4A* mutation is indicated with a red arrow. PD = Parkinson’s disease; MSA = multi system atrophy; SD = semantic dementia; EPS = extrapyramidal symptoms.

**Figure 3 biomolecules-12-00440-f003:**
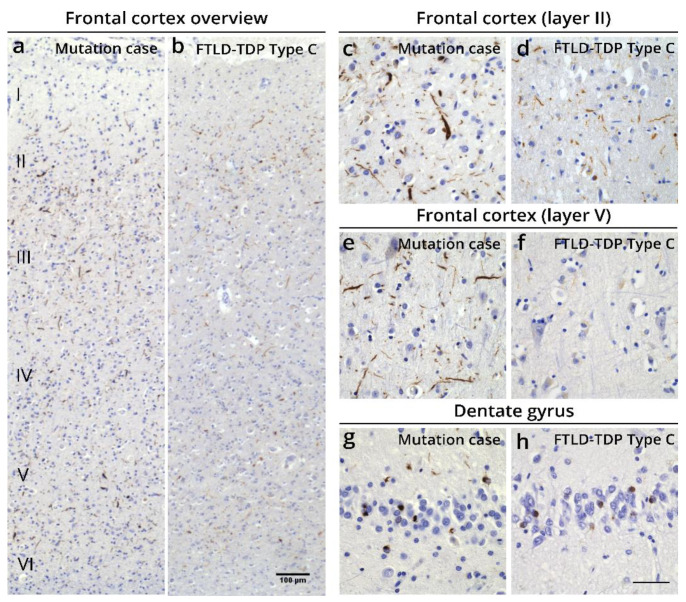
Histopathological characterization of the FTLD-TDP case with an R64Gfs*90 *TUBA4A* mutation. (**a**–**h**) pTDP-43 pathology is spread over all layers of the frontal cortex in the R64Gfs*90 *TUBA4A* mutation case, whereas in a typical FTLD-TDP type C case predominantly the second layer is affected (**a**,**b**). pTDP-43 pathology in the R64Gfs*90 *TUBA4A* mutation case mainly consists of dystrophic neurites of various length and thickness, and few neuronal cytoplasmic inclusions in the frontal cortex (layers II and V depicted; (**c**–**f**)). The dentate gyrus shows typical pTDP-43-positive cytoplasmic inclusions (**g**,**h**). Scale bars represent 100 μm (**a**,**b**) and 50 μm (**c**–**h**).

**Figure 4 biomolecules-12-00440-f004:**
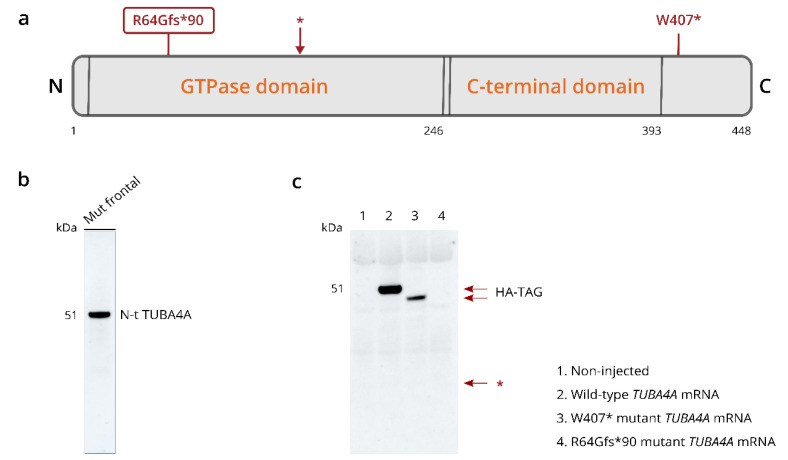
R64Gfs*90 *TUBA4A* mutation does not give rise to a mutant protein fragment. (**a**) Schematic overview of the TUBA4A protein structure indicating the location of the R64Gfs*90 and W407* mutations. The asterisk indicates the location of the early stop codon after amino acid 154 due to the R64Gfs*90 frameshift. The predicted molecular weight of the protein fragment is 16.8 kDa. (**b**) Western blot on the total fraction of the frontal cortex of the R64Gfs*90 *TUBA4A* mutation case using an antibody directed against the N-terminal part of TUBA4A, indicating that there was no truncated TUBA4A protein product present. (**c**) Western blot using an anti-HA-tag antibody on zebrafish lysates at 6 h post fertilization after the injection of wild-type, R64Gfs*90 or W407* mutant *TUBA4A* mRNA. No R64Gfs*90 TUBA4A protein fragment could be detected around 16.8 kD (asterisk), while the W407* shortened protein product was present. N = N-terminus; C = C-terminus.

**Figure 5 biomolecules-12-00440-f005:**
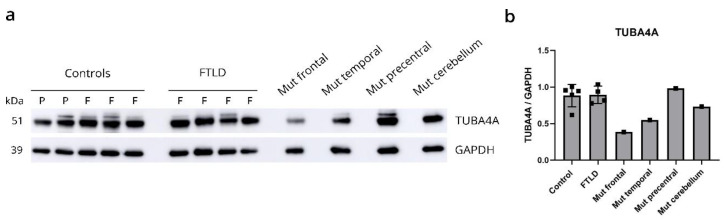
R64Gfs*90 *TUBA4A* mutation causes reduction in levels of wild-type TUBA4A protein. (**a**) Biochemical analysis of the total TUBA4A expression levels in control cases (F = frontal, P = precentral cortex), FTLD-TDP patients (F = frontal cortex) and different brain regions of the R64Gfs*90 *TUBA4A* mutation case (frontal cortex, temporal cortex, precentral cortex and cerebellum). (**b**) Quantification relative to GAPDH shows decreased TUBA4A protein expression in affected brain regions in the R64Gfs*90 *TUBA4A* mutation case. Each data point represents a single patient.

**Table 1 biomolecules-12-00440-t001:** *TUBA4A* RNA levels. Fold change of R64Gfs*90 mutant *TUBA4A* and WT (wild-type) *TUBA4A* RNA levels compared to control samples (*n* = 3) and relative to GAPDH for the frontal and temporal cortex of the R64Gfs*90 *TUBA4A* mutation case. Probes were used in combination and worked through competitive inhibition. Fc = frontal cortex, Tc = temporal cortex.

	R64Gfs*90 Case Fc	R64Gfs*90 Case Tc
Mutant TUBA4A	2.29	4.78
WT TUBA4A	0.24	0.51

## Data Availability

Most data generated or analyzed during this study are included in this published manuscript and in its Appendix A files. Additional data are available from the corresponding author upon reasonable request.

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
