# Peer review of "Frontotemporal Lobar Degeneration Case with an N-Terminal TUBA4A Mutation Exhibits Reduced TUBA4A Levels in the Brain and TDP-43 Pathology"

_biomolecules, 2022, doi:10.3390/biom12030440_

Round 1
Reviewer 1 Report
In the present study of an autopsy case with frontotemporal dementia in the presence of a previously unknown mutation in the TUBA4A gene (frame shift mutation c.187del; p.Arg64Glyfs*90), the authors analyzed neuropathological changes, allele-related TUBA4A mRNA expression, and TUBA4A protein expression at the Western blot level and at the immunohistochemical level, among others. Neuropathologically, the case was classified as FTLD-TDP most closely corresponding to FTLD-TDP type C with additional conspicuous TDP pathology in layer 5 of the neocortex. By qPCR, the mRNA of the mutant gene was detected, but no truncated protein was found by Western blot. The total TUBA4A mRNA amount was reduced as well as the total TUBA4A protein amount in the Western blot. The authors conclude that haploinsufficiency is the possible underlying pathogenetic mechanism.
Great care was taken in the characterization of this most interesting case. The manuscript text and the presentation of the results are very well done.
There are only a few minor criticisms or ambiguities/questions.
- Supplementary table 2 lists a TDP inclusion pathology in the precentral cortex. Since the patient clinically had FTD but no additional ALS, it would be interesting to know how pronounced the TDP pathology was compared to the frontal cortex? Is it possible that there was already preclinical motor system disease? In this regard, was there an analysis of the pyramidal tract regarding myelin loss or mircoglia activation? No TDP pathology was found at the level of the cervical spinal cord or in the ncl. of the hypoglossal nerve according to suppl. table 2. Deeper sections of the spinal cord (especially of lumbar segments) were probably not present. For a final neuropathological classification, the authors should add the findings on the possible degeneration of the pyramidal tract and finally assess the neuropathological changes of the motor system.
- Comments should be made on the genetics of the sibling with PD. If an identical mutation is present, this would underline the pathogenicity of the present TUBA4A mutation. Information on the genetics of the father, who also had a Parkinson-like diesease (MSA), will probably not be available.
- Figure 5 and supplementary figure 3b. Figure 5a, b shows a clear reduction of the TUBA4A protein for the mutation case in the frontal cortex, but supplementary figure 3b shows an overexpression of the protein in immunohistochemistry at least in layer 5 of the frontal cortex. What explanation do the authors have for this?
- Figure 5: Why were the Western blot samples for the frontal cortex and the precentral cortex combined for the control cases and but not for the FTLD-TDP cases?
- Fig. 5a and b. The abbreviation "mut central" should be replaced by the abbreviation "mut precentral" for better understanding.
- Suppl table 1. Is there any information on the TUBA4A mutation status of the FTLD-TDP cases as well as the control cases? In particular, the mutation status of the FTLD-TDP type C cases would be interesting.
Reviewer 2 Report
Thank you for the opportunity to review this interesting manuscript. In this study the authors present a case report of an individual affected by semantic variant of primary progressive aphasia carrying a novel TUBA4A genetic variant, R64Gfs*90. This includes neuropathological investigation, family history, genetic screening, expression analysis at the gene and protein levels, as well as in vivo modelling in zebrafish. The main findings of this study is that this novel N-terminal mutation causes haploinsufficiency of TUBA4A, which is demonstrated at the protein and RNA levels by western blot analysis and qPCR of patient neuronal tissue, respectively. This finding is then replicated in zebrafish model. The authors further report differences in the neuropathology of this patient compared with other FTLD cases, and highlight the association of TUBA4A N-terminal mutations with non-ALS phenotypes. The findings of this study are indeed of interest to the neurodegenerative diseases research community, however I have a few minor queries that require some clarification in the manuscript.
Specific comments
Pg 2 Line 53-54 – This sentence suggests that 30% of FTD cases have a recessive genetic cause of disease – was that the intention? Perhaps worth revising to instead include the percentage of cases explained by known mutations.
Section 2.1 – Mutation status – can you please state whether other known FTD mutations/genes had been ruled out in this patient and how? Can you please also comment on the mutation status of the additional four FTLD-TDP cases, and whether they carried any causal mutations (eg. a C9orf72 expansion) and whether they were screened for TUBA4A mutations? These details should also be added to Supp. Table 1.
Section 2.2, Pg 2 Line 87 – Unclear why Table 1 is referenced here in the methods when this appears to be a results table.
Section 2.4 – Was IF also performed for controls and the additional FTD cases?
Section 2.7 – Please briefly refer to the purpose of the W407* construct.
Section 3 Pg 4 – Were DNA samples from any other affected family members available for screening of the TUBA4A mutation to establish segregation? Particularly the living affected sibling. This result would be useful to discuss the overlap/difference between FTD and PD.
Figure 3 – Z-score colour coding key is pixelated, is it possible to increase image resolution?
Section 3 Pg 7 Line 244-246 – Could you include more representative images in Figure 3 to demonstrate that this is a “prominent feature”?
Section 3, Pg 7 Line 253-256, Figure 4b– Can you specify exactly which residues the N-terminal directed TUBA4A antibody actually binds? Does it bind upstream of the mutation site? If not how can you be sure there is truly no protein present, rather than a failure of binding due to the change in sequence? If not, please address this in the discussion as a caveat and consider repeating these experiments with an antibody binding upstream of the mutation site.
Table 1 – Perhaps I am misinterpreting this table, however I interpret this to show a fold change of 0 compared to controls for WT TUBA4A in the FC and TC. However this is not reflected in pg7 lines 260-261. Could you please clarify.
Figure 5 – Please label which control lanes are frontal cortex and which are precentral cortex to allow proper comparison to the mutation carrier. For consistency and ease of reference please also add a label for frontal cortex for the other FTLD cases.
Section 4, line 299 – please name these genes/proteins.
